

# Using *Ipomoea aquatic* as an environmental-friendly alternative to *Elodea nuttallii* for the aquaculture of Chinese mitten crab

Linlin Shi[1,2], Meijuan Jin[2], Mingxing Shen[2], Changying Lu[2],
Haihou Wang[2], Xingwei Zhou[2], Lijuan Mei[3] and Shixue Yin[3]

[1] College of Agriculture, Yangzhou University, Yangzhou, Jiangsu, China
[2] Research Center of Agricultural Resource and Environment, Institute of Agricultural Science in Taihu Lake District, Suzhou, Jiangsu, China
[3] College of Environmental Science and Engineering, Yangzhou University, Yangzhou, Jiangsu, China

Corresponding authors
Mingxing Shen, smx@jaas.ac.cn
Shixue Yin, sxyin@yzu.edu.cn

## ABSTRACT

*Elodea nuttallii* is widely used in Chinese mitten crab (CMC) rearing practice, but it is not a native aquatic plant and cannot endure high temperature. Thus, large *E. nuttallii* mortality and water deterioration events could occur during high-temperature seasons. The aim of this study was to identify the use of local macrophytes in CMC rearing practice, including *Ipomoea aquatic* and *Oryza sativa*. A completely randomized field experiment was conducted to investigate the crab yield, water quality, bacterioplankton community and functions in the three different systems (*E. nuttallii*, *I. aquatic*, and *O. sativa*). Average crab yields in the different macrophyte systems did not differ significantly. The *I. aquatic* and *O. sativa* systems significantly decreased the total nitrogen and nitrate-N quantities in the outflow waters during the rearing period compared to the *E. nuttallii* system, and the *I. aquatic* and *O. sativa* plants assimilated more nitrogen than the *E. nuttallii* plant. Moreover, the significant changes of bacterioplankton abundances and biodiversity in the three systems implied that cleanliness of rearing waters was concomitantly attributed to the differential microbial community and functions. In addition, principle component analysis successfully differentiated the bacterioplankton communities of the three macrophytes systems. Environmental factor fitting and the co-occurrence network analyses indicated that pH was the driver of bacterioplankton community structure. Functional predictions using PICRUSt (v.1.1.3) software based on evolutionary modeling indicated a higher potential for microbial denitrification in the *I. aquatic* and *O. sativa* systems. Notably, the *O. sativa* plants stopped growing in the middle of the rearing period. Thus, the *I. aquatic* system rather than the *O. sativa* system could be a feasible and environmental-friendly alternative to the *E. nuttallii* system in CMC rearing practice.

## INTRODUCTION

The Chinese mitten crab (CMC), *Eriocheir sinensis,* is considered an invasive species in Europe and North America (*Brodin & Drotz, 2014*; *Hanson & Sytsma, 2008*), but it is an expensive delicacy in Asia (*Chen & Zhang, 2007*; *Food and Agriculture Organization of the United Nations (FAO), 2019*). In 2014, 796,621 tons of farmed CMCs were produced (*Zeng et al., 2013*); crabs were primarily bred in ponds and lakes (*Zeng et al., 2013*). The mitten crabs produced in Yangcheng Lake, Suzhou, China, are of high quality and have high economic value (*Gu et al., 2013*). Most of the crabs produced in Yangcheng Lake are exported to Shanghai, Hong Kong, and high-profit foreign markets.

Aquatic plants are required for mitten crab farming. The plants provide shelter for the crabs during exuviation, which is an important part of crab growth (*Meng et al., 2013*). In addition, aquatic plants assimilate excess nutrients, improve water cleanliness, and absorb solar radiation to maintain cool water temperatures. These properties increase crab growth, yield, and quality (*Zhan & Yang, 2015*).

*Elodea nuttallii*, a perennial aquatic plant native to North America, provides these benefits, and is thus widely used in CMC aquaculture (*Wang et al., 2016*). However, *E. nuttallii* cannot withstand high temperatures (*Zhan & Yang, 2015*). In Yangcheng Lake, summer air and water temperatures typically reach 35–40 and 26–34 °C, respectively, which frequently leads to massive *E. nuttallii* die off, resulting in serious water quality deterioration (*Wu et al., 2016*). Under such conditions, crab growth is negatively affected due to loss of shelter for exuviation hiding places and poor water quality (*Zhan & Yang, 2015*). Consequently, alternative aquatic plants are required to facilitate mitten crab aquaculture in areas such as Yangcheng Lake.

Local plants that have adapted to local conditions are the best candidates for *E. nuttallii* replacement. For example, *Ipomoea aquatica* is a semiaquatic, tropical/subtropical plant that might be applicable to mitten crab aquaculture (*Zhang et al., 2014*). *I. aquatic* grows well in shallow waters, withstands high temperatures, and efficiently removes nutrients (e.g., nitrogen and phosphorous) from water bodies (*Liu et al., 2007*; *Wei et al., 2017*). Furthermore, the tender shoots and leaves of *I. aquatic* are consumable, and providing additional economic value. Alternatively, *Oryza sativa* is a submerged grain crop that is common in Asia, and its assimilation ability of nitrogen and phosphorus is similar to *I. aquatic* (*Nawaz & Farooq, 2017*). *O. sativa* is thus another candidate for *E. nuttallii* replacement.

Therefore, in this study, we aimed to answer three questions. Which of the two locally adapted plants would most adequately replace *E. nuttallii*, similarly improving crab yield and quality? How do the candidate plant systems affect water quality and the associated environmental characteristics? Considering microorganisms play important roles in nutrient conversion in wastewater (*Daims, Taylor & Wagner, 2006*), are the two plant systems associated with different bacterioplankton communities that differentially affect water quality? Answers to these questions are of critical importance to crab farmers, and those who are concerned with water quality. Despite the importance of these questions, they remain unanswered. We thus aimed to address the above questions by evaluating candidate aquatic plant systems, and exploring the associated bacterioplankton communities.

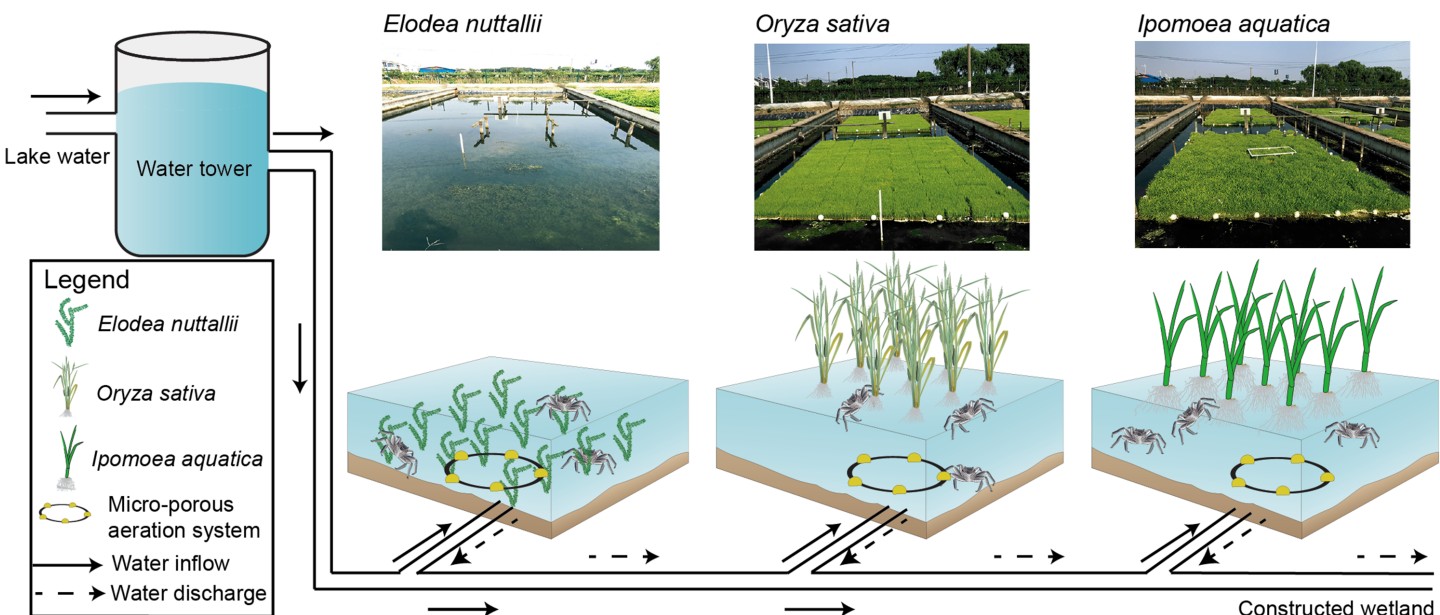

**Figure 1 Experimental design and schematics for three macrophyte systems.** *Elodea nuttallii*, *Oryza sativa*, and *Ipomoea aquatica*. Each macrophyte was planted in a separate sub-pond (three replicate ponds per species). The sub-ponds were connected by a series of PVC pipes. Pond water was replenished from the Yangcheng Lake water, and discharged water flowed into a constructed wetland. The growth of all three macrophytes was restricted to 60% of the pond areas. A micro-porous aeration system was used nightly to ensure sufficient aeration. Photo credit: Linlin Shi.

## MATERIAL AND METHODS

### Pond construction, seedling preparation, and floating system construction

All of the experiments were performed in Lianzigang, Suzhou, China (31°27′40.18″N, 120°43.5′5.32″E). This region has a subtropical monsoon climate, with an annual average rainfall of 1,076.2 mm (http://www.pmsc.cn/). Annual average temperature in Lianzigang is 15–17 °C, with high temperatures of 35–40 °C in July and August (http://www.pmsc.cn/). All of the experiments were conducted between May 1 and November 30, 2017.

The crab-rearing system used here consisted of a water inlet, a water outlet, a micro-porous aeration system, a pond, floating macrophytes, and vertical posts to anchor the macrophytes (Fig. 1). The pond, which was mechanically excavated, had a total area of 1,000 m² and was 1.5-m deep. The pond was separated into nine sub-ponds with cement walls that were 0.4 m wide and 1.5 m high. Each sub-pond had an area of ~100 m² and an independent staff gauge. Sub-pond bottoms were left in a natural state to allow crabs to burrow, and to facilitate *E. nuttallii* root establishment and growth. *I. aquatic*, *O. sativa*, and *E. nuttallii* were grown in separate sub-ponds with three replicates per plant using a completely randomized design.

Sub-ponds were disinfected using sodium hypochlorite. After disinfection, water was pumped into each sub-pond to a depth of ~10 cm. Basal fertilizer (45 kg ha⁻¹ of compound-fertilizer; N:P₂O₅:K₂O = 15:15:15) was applied 2–3 days before seedling transplantation.

*Oryza sativa* and *I. aquatic* seedlings were prepared by sowing seeds on a patented nutritional matrix (0.16 m$^2$) containing sufficient nutrients for seedling growth. The matrix had sufficient buoyancy to allow seedlings to float on the surface of the water. Seeds were germinated on the matrix, grown for 30 days in a greenhouse at 25 °C, and then transferred to the bottom of the appropriate sub-ponds. Seedling matrices were arranged side by side until seedling coverage reached 60% of the total water surface area. Additional water was then pumped into each sub-pond to increase the water depth to 20 cm, and seedlings were allowed to re-establish growth for 5–7 days. Upon growth re-establishment, water was again gradually and gently pumped into the sub-ponds to increase water depth to 1.0 m. At this depth, all of the seedlings were floating. Seedlings were fixed in place using ropes attached to buoys, which were fastened to posts in the sides of the cement dividers. *E. nuttallii* seedlings were directly transplanted from other ponds. *E. nuttallii* cluster spacing was $0.5 \times 0.5$ m, with 40 seedlings per cluster. The transplanted *E. nuttallii* plants covered ~60% of the total water surface area. Upon growth re-establishment, water was pumped into the *E. nuttallii* sub-ponds to a depth of 1.0 m.

## Crab pond management, nutrient measurements, and crab yield determination

Crab ponds were managed using standard CMC rearing methods (*Zhan & Yang, 2015*). Juvenile crabs, with an average weight of 15 g, were purchased from a local company (Su'an Fishery Co., Ltd., Nantong, China) and added to the experimental sub-ponds at a density of about 12,000 individuals per hectare. Crab feed (bait) was purchased from the Tongwei Group (http://www.tongwei.com/). The nutrient composition of the feed was varied to meet the different needs at each of the growth stages. Crabs were fed twice per day dependent on growth stage, as recommended by the bait manufacturer. Crabs were not dosed with antibiotics or chemical fishery drugs. A micro-porous aeration system was used nightly to ensure sufficient aeration. Approximately five cm of water was pumped out of all ponds each week, and replaced with an equal volume of fresh water from Yangcheng Lake (depending on local precipitation).

Outflow water samples were taken every 7–10 days during the rearing period, and immediately frozen at −20 °C. At the end of the experiment, nutrients in the outflow samples were measured using an auto analyzer (SKALAR SAN$^{++}$, Breda, the Netherlands); the nutrients measured included total nitrogen (TN), total phosphorous (TP), ammonium-N (NH$_4^+$-N), nitrate-N (NO$_3^-$-N), and nitrite-N (NO$_2^-$-N). Absolute cumulative nutrient quantity in discharged water was measured by nutrient concentration in discharged water and outflow water quantity estimated by staff gauge. Outflow pH was measured using a WTW portable pH meter (ProfLine 3310; WTW, Weilheim, Germany). Inlet water samples were taken and measured every month during the rearing period, and average nutrient concentrations were finally calculated. Dissolved oxygen (DO) concentrations were not measured, because preliminary studies identified large spatial and temporal variations in DO concentrations (Fig. S1) due to uncontrollable factors (i.e., air temperature, air pressure, water disturbances, aeration, activities of aquatic organisms, and photosynthesis of plants and algae) (*Dai et al., 2013*).

The biomass yields (TBM) of *E. nuttallii*, *I. aquatic*, and *O. sativa* were estimated by dry matter productivity in unit area (DM) and aquatic plant areas (APA) in each pond. The fresh plant tissues in one m$^2$ area were collected and oven-dried with three replicates in each pond at the end of the experiment, and average dry weight in one m$^2$ area was considered as DM. APA was equal to 60% of total area of each pond. Thus, the TBM could be calculated by the equation: TBM = DM × APA. Trimmed plant tissues were included in the biomass production estimates. As *I. aquatic* and *E. nuttallii* has a sprawling growth pattern, it was periodically cut back outside of the rope-restricted area to maintain ~60% coverage to keep constant aquatic plant coverage (Trimming details see Figs. S1 and S2). Nutrients assimilated by the plants were calculated based on plant biomass and nutrient concentrations. The tissue-mixed plant samples that were collected in different growth stages were used for nutrient concentration measurement. Table S1 showed the average nutrient concentrations of plants.

To assess crab production, mature crabs were harvested using crab traps, and remaining crabs were captured by hand at night, after the pond water was completely drained. Males and females were manually separated and weighed.

## Characterization of bacterioplankton communities

As crab rearing is sensitive to high air temperatures (*Yuan et al., 2017*), the bacterioplankton communities during periods of high temperatures were more interesting. Thus, the bacterioplankton community was assessed on July 2, 2017, when the maximum air temperature reached 37 °C. To obtain sufficient bacterioplankton biomass for community profiling via DNA sequencing, we collected 10 L of water at 20 cm depth from each replicate pond. Water samples were filtered through a 0.22-μm polycarbonate membrane (Millipore, Billerica, MA, USA). DNA was extracted from the filtered biomass using a FastDNA Spin Kit for soil (MP bio, Solon, OH, USA), following the manufacturer's protocols. Extracted DNA concentration was determined with a NanoDrop 2000 UV–vis spectrophotometer (Thermo Scientific, Wilmington, NC, USA); DNA quality was assessed with gel electrophoresis on a 1% agarose gel. The V4 hypervariable region of the bacterial 16S rRNA gene was amplified using PCR, with the primers 563F (5′-AYTGGG YDTAAAGVG-3′) and 802R (5′-TACNVGGGTATCTAATCC-3′) (*Cardenas et al., 2010*), following previously described protocols (*Wang et al., 2017*). The amplified PCR products were purified using an AxyPrep DNA Gel Extraction Kit (Axygen Biosciences, Union City, CA, USA), and quantified using a QuantiFluor-ST kit (Promega, Madison, WI, USA), following the manufacturer's instructions. Purified amplicons were pooled in equimolar concentrations and sequenced on an Illumina MiSeq platform (Illumina, San Diego, CA, USA) at Majorbio Bio-Pharm Technology Co. Ltd. (Shanghai, China), following standard protocols.

Raw sequence reads were demultiplexed using QIIME (v.1.9.1) (*Caporaso et al., 2010a*). Barcoding adapters and PCR primers were cleaved using cutadapt (v.1.16) (*Martin, 2011*). Low-quality reads were removed from the dataset with USEARCH10 (*Edgar & Flyvbjerg, 2015*), using the "fastq_filter" command with the parameters maxee = 1 and truncqual = 15. The remaining paired-end reads were merged using the

"fastq_mergepairs" command in USEARCH10 (*Edgar & Flyvbjerg, 2015*). Merged read abundances were normalized across samples by randomly subsampling 28,000 sequences from each sample. A zero-radius OTU (zOTU) table was produced from the sequence reads using the Unoise3 algorithm (*Edgar, 2018*). The taxonomic classification of each zOTU was assigned using the UCLUST algorithm against the Silva (SSU123) 16S *r*RNA database with default parameters (*Caporaso et al., 2010b*; *Edgar, 2010*).

An additional OTU table was generated to predict bacterioplankton functions. The abundance-normalized sequences were clustered at the 97% nucleotide similarity cutoff level into OTUs using the "pick_closed_reference_otus.py" function in QIIME (v.1.9.1) (*Caporaso et al., 2010b*). OTUs were taxonomically classified using the UCLUST algorithm (*Edgar, 2010*) against the Greengenes (gg_13_5) reference database (*DeSantis et al., 2006*) with default parameters. The resulting OTU table was analyzed using PICRUSt (v.1.1.3) (scripts "normalize_by_funtion.py," "predict_metagenomes.py," "categorize_by_function.py," and "metagenome_contributions.py") (*Langille et al., 2013*). The PICRUSt algorithm produced a table of functional Kyoto Encyclopedia of Genes and Genomes orthologs (KOs). To obtain OTU-specific counts of genes (*Da Fonseca et al., 2019*; *Fan et al., 2018*) associated with nitrification and denitrification, the PICRUSt script "metagenome_contributions.py" with -l option was applied to selected KOs (K00370, K00371, K00374, K02567, K02568, K00368, K15864, K04561, K02305, K00376, K10535, K10944, K10945, and K10946) (Script S1).

## Statistical analysis

Statistical analyses were primarily performed in R (v.3.3.2) (*R Core Team, 2013*). To identify significant differences among treatments, the Levene and Kolmogorov–Smirnov tests were used to check the homogeneity of variances and data normality, respectively. One-way ANOVA was used to determine the significance among the treatments, and Tukey's HSD test was then applied for multiple comparisons. If the measurement variable did not meet the normality assumption, a Kruskal–Wallis test was performed instead of one-way ANOVA. We considered $P < 0.05$ to be statistically significant, unless otherwise noted.

The bacterioplankton communities were analyzed using the vegan package (*Dixon, 2009*) in R. The matrix of zOTU abundances was transformed prior to distance-based analyses using the "decostand" function with the Hellinger method (*Legendre & Gallagher, 2001*). Principal component analyses were performed using the "rda" function in vegan to visualize differences among the bacterioplankton communities from the different macrophyte systems. Environmental variables were then fitted and projected onto an ordination using "envfit" function in vegan based on 1,000 permutations.

To investigate the correlation of microbial taxa and environmental variables, we constructed a co-occurrence network using CoNet (*Faust & Raes, 2016*), based on the zOTU table and the environmental variables. In the co-occurrence analysis, the read count matrix was first filtered, and only those zOTUs with at least seven minimum occurrence values across the nine samples were retained. Pair-wise associations among zOTUs and environmental factors were calculated using the Pearson, Spearman, Kendall, Bray–Curtis, and Kullback–Leibler correlation methods. The initial top and bottom edge

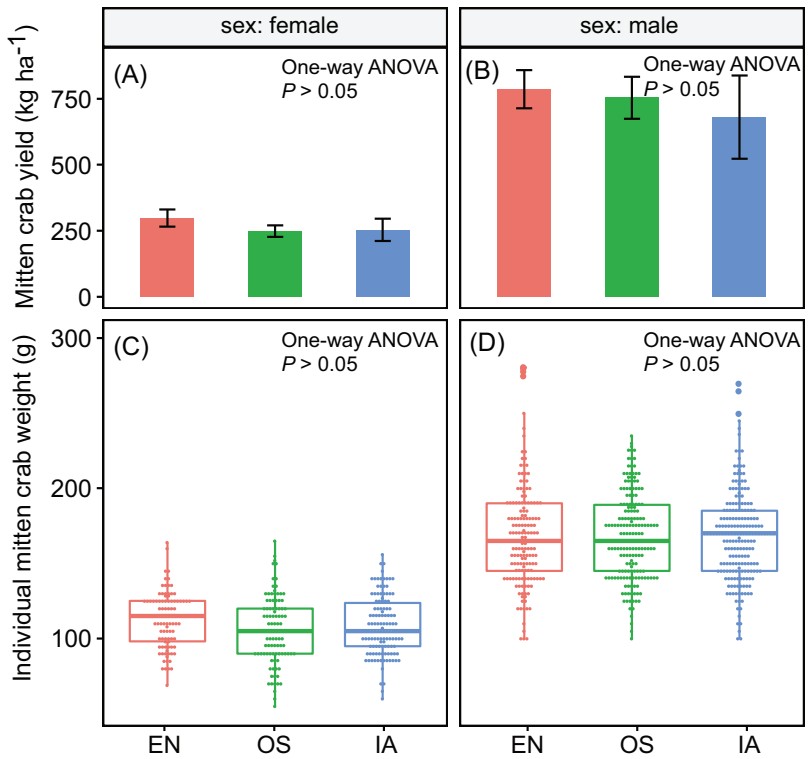

**Figure 2** **(A–D) the yield and individual weight distributions of the Chinese mitten crabs did not differ significantly among macrophyte systems.** Error bars denote the standard deviations of the mean ($n = 3$). EN, *Elodea nuttallii*; OS, *Oryza sativa;* IA, *Ipomoea aquatica*.

numbers were set at 1,000. For each edge and each measure of association, 1,000 permutation scores and 1,000 bootstrap scores were computed. The resultant networks were visualized using Cytoscape (v.3.2.1) (*Shannon et al., 2003*).

## RESULTS

### System performance, crab yield, and water quality

The macrophyte systems floated steadily throughout the whole experiment despite two moderate windstorms. Thus, the floating systems described here were suitable for plant growth on water surfaces. All of the plants grew well initially (between May and August). However, *O. sativa* began to go to seed (indicating the start of the reproductive stage) in early August, about 25 days earlier than the normal (September, if *O. sativa* is grown in a field). After early August, *O. sativa* plants yellowed and the roots darkened. In contrast, *I. aquatic* and *E. nuttallii* grew well until the end of the experiment.

Crab yields among the three macrophyte systems did not differ significantly. The average yields of male and female crabs were ~740 and ~267 kg ha$^{-1}$, respectively (Figs. 2A and 2B). Individual weight is one of the most important factors determining the value of commercial crabs. Here, the individual weight distributions did not differ significantly among plant systems ($P > 0.05$). The median weights of male and female crabs were 168 and 109 g, respectively (Figs. 2C and 2D).

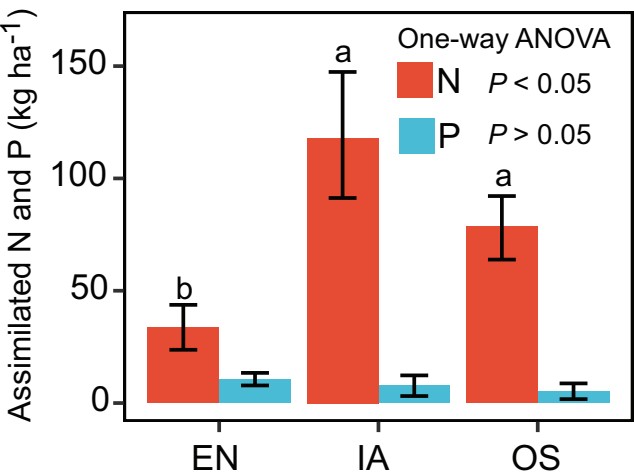

**Figure 3** Assimilation of nitrogen and phosphorous by EN, *Elodea nuttallii*; IA, *Ipomoea aquatica*; and OS, *Oryza sativa*. Different letters above bars represent significant differences among plant systems ($P < 0.05$; Tukey's HSD test). Error bars denote the standard deviations of the means ($n = 3$).

The nitrogen assimilated by *I. aquatic* was estimated to be 118 kg ha$^{-1}$, based on macrophyte biomass weight and nutrient concentrations. This level of assimilation was 3.5 times that of *E. nuttallii*, and 1.5 times that of *O. sativa* (Fig. 3). In contrast, the phosphorous assimilated by *I. aquatic*, *E. nuttallii*, and *O. sativa* did not differ significantly.

The average TN, TP, NH$_4^+$-N, and NO$_3^-$-N concentrations of inlet waters in the rearing period were 1.25 ± 0.44 (mean ± SD), 0.03 ± 0.02, 0.25 ± 0.43, and 0.57 ± 0.47 mg L$^{-1}$, respectively (Fig. 4). The TN content in 31.3% of the samples from the *E. nuttallii* system exceeded the Environmental Quality Standards for Surface Water (GB3838-2002) limit for type III water. In contrast, the TN content in 9.4% and 15.6% samples from the *O. sativa* and *I. aquatic* systems, respectively, exceeded the type III water quality limit (GB3838-2002) (Fig. 4). The number of samples that exceeded the TP and NH$_4^+$-N type III water quality limits did not differ substantially among treatments. No samples exceeded the water quality limit for NO$_3^-$-N (10 mg L$^{-1}$). Cumulative curve analysis indicated that amount of TP and NH$_4^+$-N accumulated over the course of the experiment (up to November 27) did not differ significantly among different plant systems. However, the levels of accumulated TN and NO$_3^-$-N were significantly higher in the *E. nuttallii* system than in the *I. aquatic* or *O. sativa* systems (Fig. 5). Average pH was 7.48 in the *I. aquatic* system, 8.16 in the *E. nuttallii* system, and 7.9 in the *O. sativa* system (Fig. 6).

## Bacterioplankton communities and predicted functions associated with nitrification/denitrification

The abundances of nearly all of the bacterioplankton phyla differed significantly among macrophyte systems (Fig. 7A). The abundances of *Acidobacteria*, *Chloroflexi*, *Firmicutes*, β-*Proteobacteria*, and δ-*Proteobacteria* were significantly higher in the *O. sativa* and *I. aquatic* systems as compared to the *E. nuttallii* system, while the abundances of

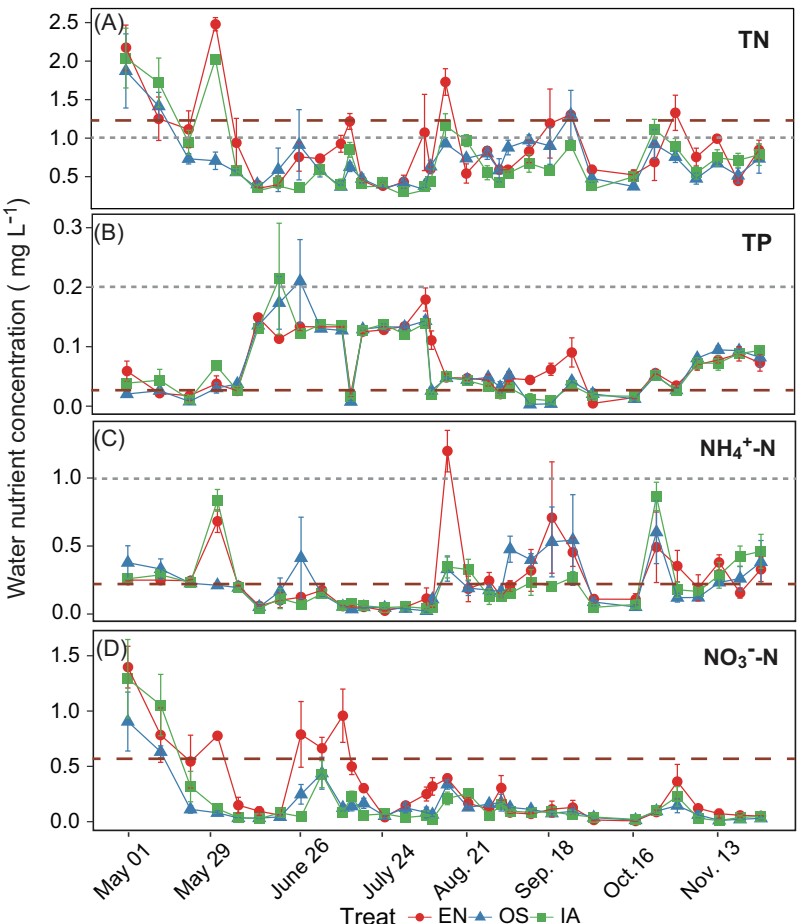

**Figure 4 (A–D) separately shows the TN, TP, NH$_4^+$-N, and NO$_3^-$-N concentrations in the rearing pond.** Error bars denote the standard deviations of the means ($n = 3$). The brown dashed lines represent the average nutrient concentrations in the inlet waters during the rearing period. The gray dashed lines represent the type III water quality limits from the Chinese Environmental Quality Standard for Surface Water (GB3838-2002). NO$_3^-$-N concentrations were all below the type III water quality limit (10 mg L$^{-1}$). EN, *Elodea nuttallii*; OS, *Oryza sativa*; IA, *Ipomoea aquatica*.

*Actinobacteria*, *Armatimonadetes,* and γ-*Proteobacteria* were significantly lower. OTU-based diversity (α-diversity) was higher in the *O. sativa* and *I. aquatic* systems than in the *E. nuttallii* system (Fig. 7B), suggesting that bacterioplankton communities were more complex in the *O. sativa* and *I. aquatic* systems than in the *E. nuttallii* system.

Principal component analyse indicated that the bacterioplankton communities were discrete among the different plants, with the first two axes explaining 83.67% of the total variation (Fig. 8A). After fitting environmental variables (Table S2) and bacterioplankton community ordination, we found that pH was significantly associated with community differences ($P < 0.05$) (Fig. 8A). The co-occurrence network analysis also identified pH as the only environmental factor that co-varied with certain taxa. These pH-correlated taxa included zOTUs belonging to the phyla *Acidobacteria*, *Chloroflexi*, *Firmicutes*, *Proteobacteria,* and *Planctomycetes* (Fig. 8B). The total average abundance of

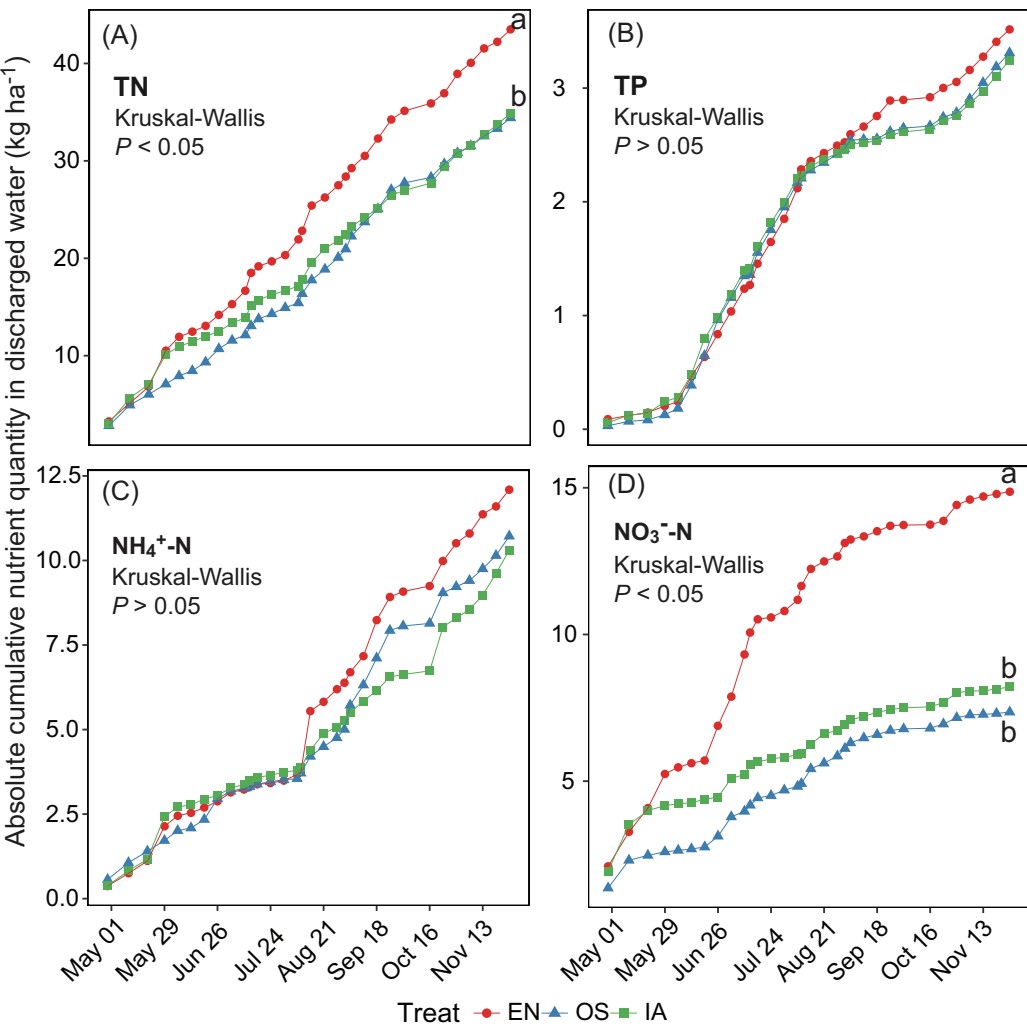

**Figure 5 (A–D) separately shows the absolute cumulative TN, TP, NH$_4^+$-N, and NO$_3^-$-N quantities in rearing period.** The statistical significance of differences in cumulative nutrient quantity among macrophyte systems was determined with a Kruskal–Wallis test ($n = 3$) at the end of the rearing periods (November 27). Different lowercase letters represent significant differences ($P < 0.05$; Tukey's HSD test). EN, *Elodea nuttallii*; OS, *Oryza sativa*; IA, *Ipomoea aquatica*.

co-occurring zOTUs was 7.7%, suggesting that pH influenced the relative abundance of the most abundant members of each community.

Potential functions of the bacterioplankton communities were predicted using PICRUSt. We focused specifically on functional genes associated with nitrification and denitrification, as these processes are highly related to nitrogen cycling and are likely to affect nitrogen concentration in pond water. Genes associated with all of the steps of denitrification (*napA*, *napB*, *nirK*, *norB*, *norC*, and *nosZ*; *Kanehisa et al., 2017*) were generally more abundant in the *I. aquatic* and/or *O. sativa* systems than in the *E. nuttallii* system. However, the abundances of *narI*, *narG*, and *narH*, which are only involved in the reduction of nitrate to nitrite (*Kanehisa et al., 2017*), were lower in the *I. aquatic* and/or *O. sativa* systems (Fig. 9). The abundances of the nitrifying genes *pmoA-amoA*,

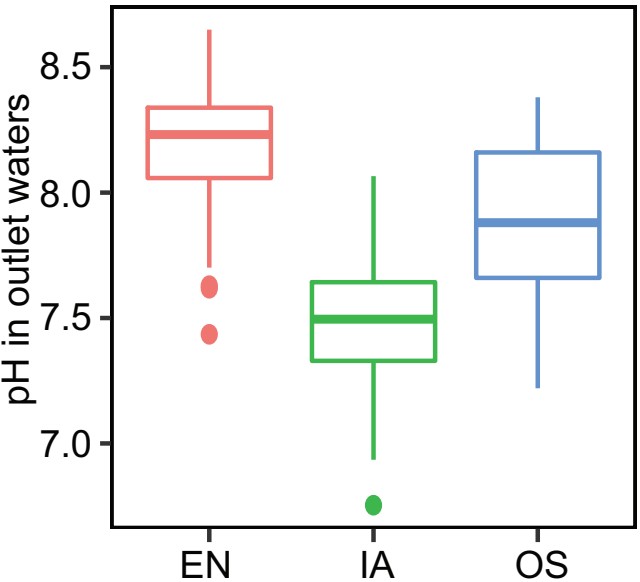

**Figure 6  pH of the outflow water samples during the rearing period.** EN, *Elodea nuttallii*; OS, *Oryza sativa*; IA, *Ipomoea aquatica*.                   

*pmoB-amoB,* and *pmoC-amoC,* which are involved in the oxidation of ammonia to hydroxylamine (*Kanehisa et al., 2017*), were significantly higher in the *O. sativa* system than in the *I. aquatic* and *E. nuttallii* systems. The abundance of *hao* was not significantly different among the three macrophyte systems, although the abundances of *narG* and *narH* were lower in the *I. aquatic* and *O. sativa* systems than in the *E. nuttallii* system (Fig. 9).

## DISCUSSION

The macrophyte system described here floated steadily throughout the whole experiment. In addition, mitten crab yield and quality did not differ significantly among plant systems (Fig. 2). Thus, our results demonstrated that local plants were a feasible alternative to *E. nuttallii*. Indeed, our data indicated that *I. aquatic* was be the best local replacement for *E. nuttallii*, as this plant grew well in ponds and provided sufficient shade. In contrast, *O. sativa* is not a suitable alternative to *E. nuttallii*, as it stopped growing in the middle of the experimental period. The mechanisms underlying the different performances of *O. sativa* and *I. aquatic* were related to the different growth behaviors and nutrient requirements of the two species. *O. sativa* has several distinctive growth stages, including tillering, heading, ripening and et al. (*Zhang et al., 2018*), which typically require different levels of nutrients (*Fairhurst et al., 2007*). For example, during the tillering stage, *O. sativa* biomass increases much faster than during other stages of *O. sativa* growth, and consequently requires a more intensive nutrient supply (*Nawaz & Farooq, 2017*). If the nutrient supply needs are not met (i.e., due to low nutrient concentrations in the water), many tillers become non-productive, and the few remaining productive tillers enter the reproductive stage earlier than normal (*Nawaz & Farooq, 2017*). In submerged paddy fields, the intensity of nutrient supply is manipulated by top-dressing fertilizers (*Fairhurst et al., 2007*). However, this technique cannot be used in crab-rearing ponds because it would pollute the water.

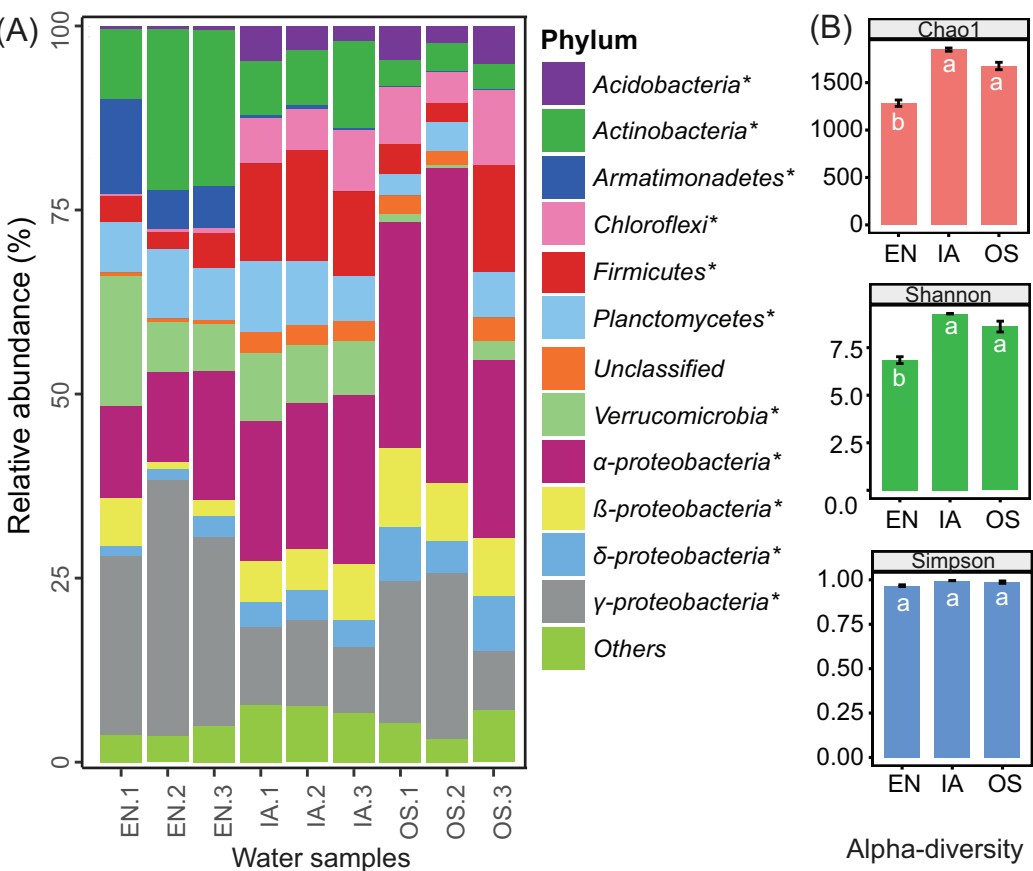

**Figure 7 Bacterioplankton abundances and diversity indices of the three macrophytes systems.**
(A) The relative abundances of the dominant bacterial phyla in the bacterioplankton communities
and (B) the associated alpha diversity. Asterisks indicate significant differences in abundances among the
macrophyte systems ($P < 0.05$ level; one-way ANOVA; $n = 3$). Error bars denote the standard errors of
the means. Different letters above bars represent significant differences among treatments ($P < 0.05$;
Tukey's HSD test; one-way ANOVA; $n = 3$). EN, *Elodea nuttallii*; OS, *Oryza sativa*; IA, *Ipomoea
aquatica*.

Unlike *O. sativa*, *I. aquatic* does not exhibit obvious physiological differences among
growth stages. *I. aquatic* has sprawling growth, with biomass increasing over rearing time
(from May to November) (*Shaltout, Al-Sodany & Eid, 2010*). Thus, low concentrations
of nutrients in the water are sufficient. We consequently concluded that *I. aquatic* is
an attractive macrophytic alternative to *E. nuttallii* in crab-rearing ponds.

Based on the Chinese national water quality standards (GB3838-2002) (*State
Environmental Protection Administration (SEPA), 2002*), 31.3% of the outflow samples
from the *E. nuttallii* system contained TN levels in excess of limits for type III waters
(Fig. 4). Thus, the outflow from ponds using the *E. nuttallii* system is likely to pollute
the downstream environment. Fewer outflow samples from the *I. aquatic* and
*O. sativa* systems, as compared to the *E. nuttallii* system, had TN levels above the type III
water quality limits. These data suggest *I. aquatic* or *O. sativa* systems would generate
less environmental pollution than *E. nuttallii* systems. This suggestion was reinforced
by the absolute cumulative nutrient quantities in outflow samples (Fig. 5), which showed

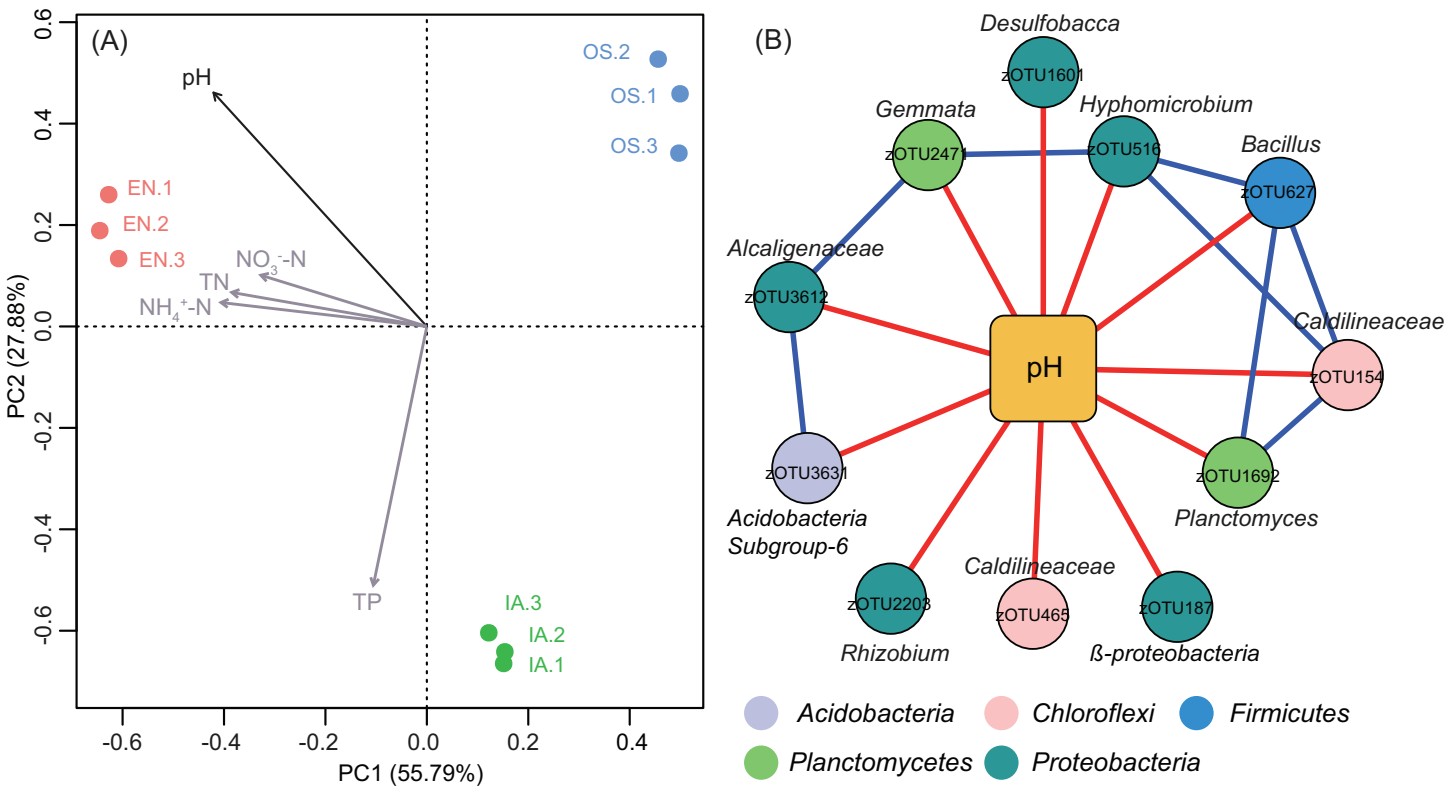

**Figure 8 Bacterioplankton community structure and environmental factors.** (A) Principal component analysis (PCA) of bacterioplankton community composition and environmental variables. The significant environmental variable ($P < 0.05$) is shown with a black arrow; other factors are shown in gray. EN, *Elodea nuttallii*; OS, *Oryza sativa*; IA, *Ipomoea aquatica*. (B) Subnetwork showing the correlation between bacterial taxa and pH. Squared nodes correspond to environmental parameters and circle nodes correspond to zOTUs. Circle nodes not assignable to genus are labeled with the names of higher taxonomic ranks, and node colors represent phyla. Red and blue colors of edge represent negative and positive correlations, respectively.

that the TN and $NO_3^-$-N concentrations were significantly higher in the *E. nuttallii* system, as compared to the *I. aquatic* and *O. sativa* systems. Indeed, *I. aquatic* and *O. sativa* assimilated more nitrogen from the water than did *E. nuttallii* (Fig. 3). Thus, it was possible that *I. aquatic* and *O. sativa* more effectively improved water quality than *E. nuttallii*, which is a desirable property for macrophytes grown in crab-rearing ponds. An alternative explanation was that the differences in water quality among plant systems were a result of the activities of macrophyte-specific bacterioplankton communities.

Bacterioplankton communities differed significantly among the three macrophyte systems (Fig. 6), possibly because of the different root deposits produced by the three plants. Terrestrial and macrophytic plants release unique root exudates that drastically alter bacterioplankton community structure (*Baudoin, Benizri & Guckert, 2003*; *Casamatta & Wickstrom, 2000*; *Nelson et al., 2013*; *Tanaka et al., 2012*; *Zhao et al., 2013*). Bacterioplankton community composition might also be regulated by pH, as pH was the only environmental factor that was significantly associated with the relative abundances of specific bacterioplankton taxa (Fig. 8). These results were consistent with previous reports, which indicated that pH affects the community structures of both terrestrial bacteria (*Fierer & Jackson, 2006*; *Lauber et al., 2009*; *Rousk et al., 2010*) and

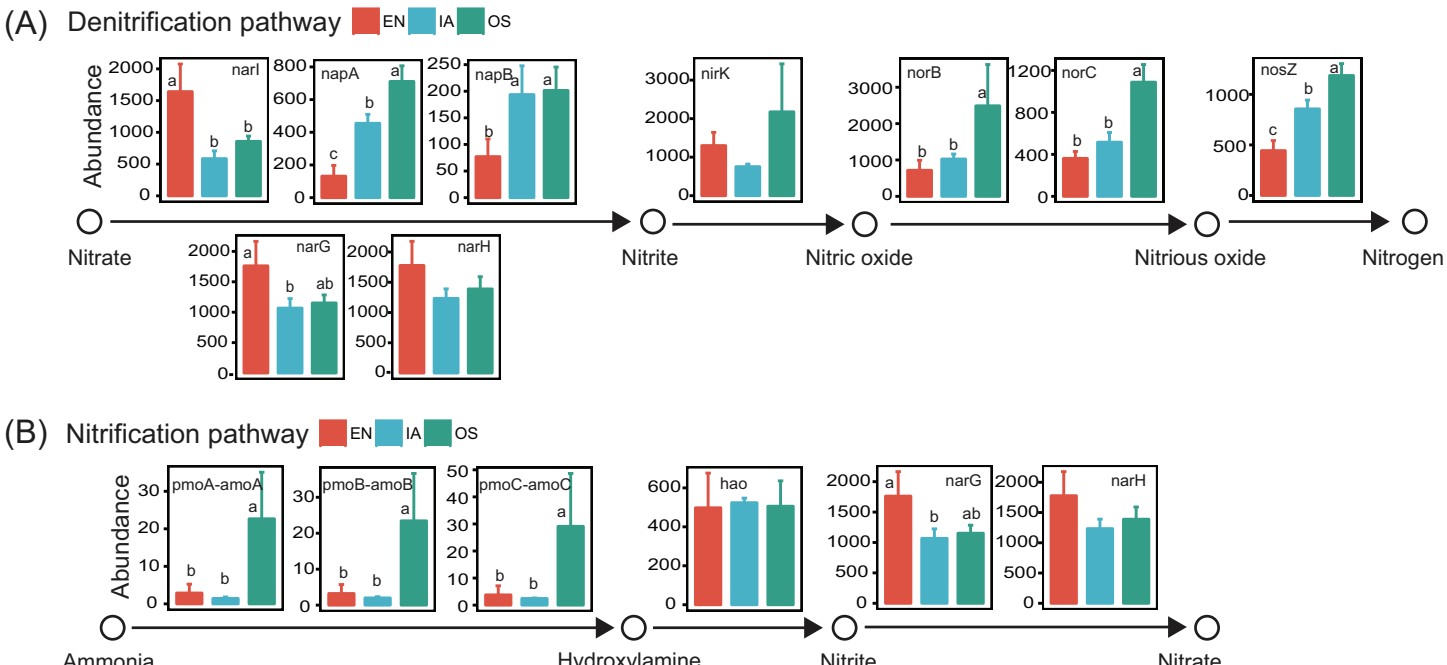

**Figure 9** **The abundances of the bacterioplankton genes associated with nitrification and denitrification among macrophyte systems, as predicted by PICRUSt.** (A) and (B) shows the denitrification and nitrification predicted gene abundances, respectively. Error bars denote the standard deviations of the means ($n = 3$). Different lowercase letters above bars represent statistically significant differences ($P < 0.05$; one-way ANOVA followed by Tukey's HSD tests). EN, *Elodea nuttallii*; OS, *Oryza sativa*; IA, *Ipomoea aquatica*.

bacterioplankton (*Ren et al., 2015*). Moreover, the significant increases in bacterioplankton diversity with lower water nutrients in the *I. aquatic* and *O. sativa* treatments (Figs. 5 and 7B) indicated that biodiversity helped improve water quality, as observed in other aquatic systems (*Cardinale, 2011*; *Gregoracci et al., 2012*).

The predicted functions of the bacterioplankton community also differed among the macrophyte systems, particularly those related to nitrification and denitrification. Most denitrifying genes, especially those associated with the reduction of nitrite, nitric oxide, and nitrous oxide, were more abundant in the *I. aquatic* and *O. sativa* bacterioplankton communities, as compared to the *E. nuttallii* bacterioplankton community (Fig. 9). Moreover, abiotic environmental factors usually control the denitrification process, that is, pH, temperature, and organic carbons (OC). The *I. aquatic* and *O. sativa* systems that could provide labile OC derived from root exudates (*Wang et al., 2018*; *Xu et al., 2008*; *Zhu & Cheng, 2011*) favored denitrification. Overall, the biotic and abiotic influences on denitrification in the *I. aquatic* and *O. sativa* systems were thought to be co-occurring, consistent with the conclusion in riparian wetlands (*Xiong et al., 2017*). In addition, concentrations of $NO_3^-$-N (and TN) were lower in the *I. aquatic* and *O. sativa* systems than in the *E. nuttallii* system. Thus, the *I. aquatic* and *O. sativa* systems might have a higher denitrification potential than the *E. nuttallii* system. Meanwhile, there were no obvious differences in the abundance patterns of nitrification-associated genes among macrophyte systems. The genes responsible for the oxidation of ammonium to hydroxylamine (*pmoA-amoA, pmoB-amoB,* and *pmoC-amoC*) were significantly more abundant in the *O. sativa*

system (but not the *I. aquatic* system) as compared to the *E. nuttallii* system, but the abundances of these genes were relatively low across all of the plant systems, with only ~2–20 copies per sample. In addition, the gene abundances in nitrification process usually could not accurately predict nitrification potential, which can be affected by other abiotic factors (*Francis et al., 2005*; *Rocca et al., 2015*; *Yao et al., 2018*). Thus, nitrification potential may not vary significantly among these three macrophytes, and further studies are required to validate these coupled nitrification–denitrification processes in future.

## CONCLUSIONS

*Elodea nuttallii* is routinely cultivated in ponds used for mitten crab aquaculture. However, *E. nuttallii* temperature sensitivity often leads to plant deterioration and decreased water quality at high ambient temperatures, negatively affecting crab production. Here, we successfully designed a floating system to support the growth of *I. aquatic* and *O. sativa* on the surfaces of crab-rearing ponds. We then compared the crab yield, outflow water quality, and bacterioplankton communities among ponds with *E. nuttallii*, *I. aquatic*, and *O. sativa* macrophyte systems. Our results indicated that *I. aquatic* growth behavior was preferable to that of *O. sativa*. Crab yields did not differ significantly among systems. Moreover, outflow water quality, as indicated by TN and $NO_3^-$-N concentrations, was better in the *I. aquatic* and *O. sativa* systems than in the *E. nuttallii* system, due to the greater nitrogen assimilation of *I. aquatic* and *O. sativa* as compared to *E. nuttallii*. In addition, the microbial communities associated with *I. aquatic* and *O. sativa* had a greater denitrification potential than the microbial community associated with *E. nuttallii*. Thus, our results indicated mitten crabs could be successfully reared using native aquatic plants. Specifically, *I. aquatic* was a suitable and environmental-friendly replacement for *E. nuttallii*, but *O. sativa* was not.

## ACKNOWLEDGEMENTS

Authors would like to thank the anonymous reviewers and Kiran Liversage for their good suggestions, which have helped us to improve this manuscript. We also thank LetPub for its linguistic assistance during the preparation of this manuscript.

### Funding

This work was supported by grants from the National Key Technology Research and Development Program of China (no. 2012BAD14B12-03), the Exploratory and Paradigm-shifting Innovation Project of Jiangsu, China (no. ZX(17)2001), and the Agricultural Science and Technology Demonstration Project of Suzhou, China (no. SNG201645). The funders had no role in study design, data collection and analysis, decision to publish, or preparation of the manuscript.

### Grant Disclosures

The following grant information was disclosed by the authors:
National Key Technology Research and Development Program of China: 2012BAD14B12-03.

Exploratory and Paradigm-shifting Innovation Project of Jiangsu, China: ZX(17)2001. Agricultural Science and Technology Demonstration Project of Suzhou, China: SNG201645.

## Competing Interests

The authors declare that they have no competing interests.

## Author Contributions

- Linlin Shi conceived and designed the experiments, performed the experiments, analyzed the data, contributed reagents/materials/analysis tools, prepared figures and/or tables, authored or reviewed drafts of the paper, approved the final draft.
- Meijuan Jin performed the experiments, contributed reagents/materials/analysis tools, prepared figures and/or tables, approved the final draft.
- Shixue Yin authored or reviewed drafts of the paper, approved the final draft.
- Changying Lu performed the experiments, approved the final draft.
- Haihou Wang performed the experiments, approved the final draft.
- Xingwei Zhou performed the experiments, approved the final draft.
- Lijuan Mei contributed reagents/materials/analysis tools, approved the final draft.
- Mingxing Shen conceived and designed the experiments, authored or reviewed drafts of the paper, approved the final draft.

## DNA Deposition

The following information was supplied regarding the deposition of DNA sequences:

Demultiplexed sequences and metadata are available from the NCBI Sequence Read Archives (SRA) under accession number SRP136316 and Bioproject number PRJNA445380.

## Data Availability

The raw measurements are available in Files S1–S5.

## Supplemental Information

Supplemental information for this article can be found online at http://dx.doi.org/10.7717/peerj.6785#supplemental-information.

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
