# Peer review of "Using Ipomoea aquatic as an environmental-friendly alternative to Elodea nuttallii for the aquaculture of Chinese mitten crab"

_PeerJ, doi:10.7717/peerj.6785_

## Round 0.1 · original submission · Major Revisions

The reviewers have commented on your above paper. They indicated that it is not acceptable for publication in its present form.
However, if you feel that you can suitably address the reviewers' comments (included below), then I invite you to revise and resubmit your manuscript.

Reviewer 1 ·

Basic reporting

no comment

Experimental design

no comment

Validity of the findings

no comment

Additional comments

This paper aimed to identify an alternative macrophyte system for Chinese mitten crab aquaculture. The authors found that the Ipomoea aquatic (IA) and Oryza sativa (OS) systems might both be suitable alternatives for Chinese mitten crab aquaculture, as both systems had similar crab yields to the Elodea nuttallii (EN) system, as well as a greater level of nitrogen removal. Overall, the manuscript is well written and the results are persuasive. I think this manuscript is worthy of publication in PeerJ. However, I have some minor comments and suggestions that should be considered by the authors.
1. The abstract is a little long and can benefit from being more concise. The less important contents may be deleted from the abstract, such as the method to estimate plant biomass.
2. Line 53: Three replicates are not enough to support your conclusion. Five or more replicates are needed in field experiments.
3. Lines 166 and 224: This content has been described twice and should be deleted in discussion section.
4. Lines 228-229: The authors stated that OS is not a suitable alternative to EN. However, in abstract, they also found that the IA and OS systems might both be suitable alternatives for Chinese mitten crab aquaculture. Please keep consistent throughout the manuscript.
5. Line 240: Why plant biomass increased linearly? Please give a reference for this sentence.
6. Lines 269-270: The authors reported that the IA and OS systems might have a higher denitrification potential than the EN system, because concentrations of NO3 were lower in the IA and OS systems than in the EN system. I cannot agree with this view, because denitrification is influenced by a number of environmental and biotic factors, including both carbon and nitrogen availability. Please refer to some published paper, such as Xiong et al. Environ. Sci. Technol., 2017: 5483–5491.
7. Lines 275-276: Nitrification potential is controlled mainly by environmental factors, although nitrification is a microbial-driven process. In many cases, gene abundance has no significant relationship with nitrification potential. Therefore, the authors should discuss this result carefully based on previous studies, such as Yao et al. RSC Advances, 2018, 8, 1875-1883.

Reviewer 2 ·

Basic reporting

This paper studied the water quality and the bacterioplankton community among three kinds of earth ponds for Chinese mitten crabs with different macrophyte. Overall this is an interesting study. However, many issues must be clarified before its publication. The introduction provides insufficient background. I don’t think the conclusions were well supported by the results.

Experimental design

'no comment'

Validity of the findings

'no comment'

Additional comments

English should be improved dramaticly. I'm not a native English-speaker, but I can find many improper expressions, such as L 8 are high quality -> are of high quality

Waterweeds -> The weed means useless. I thought aquatic plants would be better.

Second, the tile is about mitten crab culture, but the paper focuses more on water quality, and ignored the culture itself. How about the economy? are the systems economically feasible?

Third, what kinds of materials are used for the patented matrix? I see plastics in the photos, that will produce microplastics, a kind of ECs, which is harmful.

Forth, EN is a cryophilic one, but IA and OS are thermophile. The experiments were conducted mainly in summer, so it 'unfair' to EN to compare the nutrition assimilation.

Finally, the bacterioplankton was only sampled once. That is not enough to a scientific research.

Specific comments:

Title Please re-organized to make it more specific. novel ->what kind of new? alternative to what?

L18 How about the water temperature?

L21 how about other submerged macrophytes such as eel grass?

L29 functionally: what kind of function?

L50 How about the arrangement of the three groups? Did the three duplications of one group close to each other or separated by other groups?

L67 EN hill? what does it mean?

L87 I thought DO is very important. You can compare it in different ponds in at the same time. The floating system will evidently decrease the DO.

L103 Were the bacterioplanktons only sampled on one date? How about the biomass? If there's only relative abundance, the data will contribute little to the main topic.

L178 Where are the assimilated nitrogen from? You mentioned that the patented nutritional matrix containing sufficient nutrients.

L182 NO2-N was not detectable. I thought it can be deleted from the paper.

L245 How about TN in the downstream and in the inlet?

L248 I don't think these ponds are sources of pollution. I expect the water quality of inflows.



Figures:

Figure 2 L144 indicated there's no one-way ANOVA, but the figure displayed.

Figure 4 The water quality of inlet is expected.

Figure 5 What's the unit for vertical axis?

·

Basic reporting

I think the basic reporting is excellent.

Experimental design

In general, the experimental design appears robust. My main comment about experimental design concerns just one aspect - the biomass estimates used to measure assimilated nutrients. I had trouble following how biomass was estimated, and what was the purpose/implication of the trimming that was done. For example, it is stated that 1m2 of plant material was collected; does this mean all plant material fitting into 1m2, or all the material within 1m2 of area in the ponds? I also do not understand the reason for the trimming of the plants, nor how exactly this trimming was done (e.g. was it done from all plants throughout the ponds, or just the plants at the edges of the experimental areas?). If the trimming was only done at the edges, how exactly did it act to keep a 60% plant cover (line 93) throughout the experimental area? It is good that “Trimmed plant tissues were included in the biomass production estimates” (line 94), but then the biomass estimates/assimilated nutrient measurements presented here only relate to situations where the growth has been modified by trimming, which may not happen in areas where the actual aquaculture is occurring. Overall, some more clarification about the information presented in paragraph lines 91-96 would be helpful.

Validity of the findings

I think the findings will be highly valid and useful for aquaculture and environmental management applications.

Additional comments

Firstly, I commend the authors for the excellent diversity of measurement types they took, which considered together produce an effective understanding of the system. Below are some minor changes that may improve the manuscript:

1) One aim was to find a local plant species to use during aquaculture, because the previously used Elodea nuttallii is not native and has large mortality events during high temperature - this is not highlighted in the abstract. I think doing this would improve the abstract.
2) line 24: add "that" between "plant" and "might"
3) line 34: Change "In consider" to "Considering"
4) Line 61: how was this 60% measured?
5) Line 67: This mention of "hills" was unexpected. Please provide some background about this.
6) Line 83: change “was” to “were”
7) Line 96: I presume that here is referring to nutrient concentrations in the plant tissues - if this is correct, could you please state where the nutrient concentrations were obtained from.
Line 136-139: The quantification of nitrification and denitrification genes is an important aspect of the analysis but the method is only discussed in one sentence. Maybe this could be expanded, and especially a reference to another study or methods paper in which this has previously been done would be helpful.
8) Line 160: The list here contains many correlation types. I think readers will be curious to know why so many were simultaneously used? (i.e. why not just choose the one most effective?)
9) Line 231-232: please clarify this sentence.
10) Line 236: change to "enter"
11) line 258: add "that" between "exudates" and "drastically"
12) Fig. 2 legend: These error bars seem quite small - please double check that they are really Standard Deviation and not Standard Error. Also for other bar graphs.
13) Fig. 6: pH seems to drive differences in bacterial communities among treatments (Fig. 8), so I think it may be important that a formal comparison (e.g. ANOVA) is done for this variable (it does not seem like any comparison was done, only unanalysed means were described, line 192).
14) Fig. 7b: I think there could be more discussion about this result – e.g. what are the implications for an aquaculture system to have high or low microbial diversity? Also, are these graphs showing averages, and thus should have error bars?
15) Fig. 8b: It may be useful to have some more explanation of what the graph represents. Most is self-evident, but it would be good to know details such as: if there are links between two parts of the graph, does that mean the correlation is significant? And I guess the width of the linkage relates to a strength of correlation value, such as R2? Details such as this in the figure legend would be helpful.

---

## Round 0.2 · Minor Revisions

The reviewers have commented on your above paper. They indicated that the paper still needs some minor modifications. Best regards.

Reviewer 1 ·

Basic reporting

no comment

Experimental design

no comment

Validity of the findings

no comment

Additional comments

The revised manuscript has addressed my questions well. I think it can be published.

·

Basic reporting

No comment

Experimental design

No comment

Validity of the findings

No comment

Additional comments

All the changes I suggested previously have been made. I have only a few more minor changes/suggestions, mostly related to small corrections to the language (the line numbers are from the tracked changes manuscript file).

Line 25-28: this sentence is a bit confusing, maybe “and” should be added between “waters” and “were”.

Line 32: Please add a mention here about what PICRUSt is.

Line 46: change to “crab”

Line 68: as “is” between “phosphorus” and “similar”

Line 78: change to “address”

Line 108: change to “pond”

Line 128: remove “~”

Line 156: add “were” after “that”

Line 227: I could not work out why P values were Bonferroni-adjusted. Please state here the reason for this adjustment

Line 259: is it meant to be “inlet” here instead of “inflow”? The figure listed here (line 261) shows differences among treatments, but I would hope all the “inflow” is the same among treatments (i.e. the same water supply flowing into treatments with the same initial nutrients), and only after being in the treatments there are differences. Either use a correct term instead of “inflow” (line 259) or please clarify exactly what is meant by this.

Line 270: switch positions of words “different” with “among”

Line 367: is this meant to be “denitrifiers are heterotrophic”?

Line 368: add “the” after “control”

Line 385: change “this” to “these”

---

## Round 0.3 · accepted · Accept

Dear authors,

Thank you very much for improving your manuscript according to all the suggestions given. Congratulations,

#